# TiNbN Hard Coating Deposited at Varied Substrate Temperature by Cathodic Arc: Tribological Performance under Simulated Cutting Conditions

**DOI:** 10.3390/ma16134531

**Published:** 2023-06-22

**Authors:** Juan Manuel Gonzalez-Carmona, Claudia Lorena Mambuscay, Carolina Ortega-Portilla, Abel Hurtado-Macias, Jeferson Fernando Piamba

**Affiliations:** 1CONAHCYT-Centro de Ingeniería y Desarrollo Industrial (CIDESI), Av. Playa, Av. Pie de la Cuesta No. 702, Desarrollo San Pablo, Santiago de Querétaro 76125, Mexico; iortega@posgrado.cidesi.edu.mx; 2Facultad de Ciencias Naturales y Matemáticas, Universidad de Ibagué, Carrera 22 Calle 67, Ibagué 730002, Colombia; claudialorena0524@gmail.com; 3Centro de Investigación en Materiales Avanzados, S.C., Laboratorio Nacional de Nanotecnología, Miguel de Cervantes 120, Complejo Industrial Chihuahua, Chihuahua 31109, Mexico; abel.hurtado@cimav.edu.mx

**Keywords:** wear, arc PVD, tribology, adhesion

## Abstract

This study focused on investigating the adhesion and tribological properties of niobium-doped titanium nitride (TiNbN) coatings deposited on D2 steel substrates at various substrate temperatures (Ts) under simulated cutting conditions. X-ray diffraction confirmed the presence of coatings with an FCC crystalline structure, where Nb substitutes Ti atoms in the TiN lattice. With increasing Ts, the lattice parameter decreased, and the crystallite material transitioned from flat-like to spherical shapes. Nanoindentation tests revealed an increase in hardness (H) with Ts, while a decrease in the elastic modulus (E) resulted in an improved elastic strain limit for failure (H/E) and plastic deformation resistance (H^3^/E^2^), thereby enhancing stiffness and contact elasticity. Adhesion analysis showed critical loads of ~50 N at Ts of 200 and 400 °C, and ~38 N at Ts of 600 °C. Cohesive failures were associated with lateral cracking, while adhesive failures were attributed to chipping spallation. The tribological behavior was evaluated using a pin-on-disk test, which indicated an increase in friction coefficients with Ts, although they remained lower than those of the substrate. Friction and wear were influenced by the surface morphology, facilitating the formation of abrasive particles. However, the absence of coating detachment in the wear tracks suggested that the films were capable of withstanding the load and wear.

## 1. Introduction

The surface of a material refers to the external part of a body that comes into contact with the environment, where wear and corrosion can occur [1]. These phenomena pose significant challenges for industry as components of industrial machinery can be damaged, resulting in production stoppages for the replacement of worn-out and corroded components, leading to losses in time and money [2,3]. To address these challenges, industries seek continuous improvement and production time reduction employing new durable materials that optimize production processes effectively and efficiently [3,4,5]. In this regard, the deposition of thin-film has become increasingly popular in recent years due to the high demand generated in the industry. These coatings enhance the surface properties of the substrate, providing high hardness, low coefficient of friction, resistance to wear and corrosion especially in chemically aggressive environments, thereby increasing its lifespan and expanding its range of applications [2,3,4,5,6].

Adhesion is one of the most critical properties of a coating/substrate system as it determines the life cycle of the film. Various qualitative and quantitative techniques are available to assess adherence. Qualitative techniques involve determination of the behavior of surface cracks generated by an indentation, while quantitative methods seek to obtain the critical loads, both adhesive and cohesive, using dynamic scratching [7]. However, adhesion depends on intrinsic parameters, such as the relationship between the film and substrate hardness, the interface and diffusion type, morphology, and roughness. These factors make adhesion evaluation complex, requiring in-depth quantitative and qualitative analysis of the failure mechanisms of surfaces [8]. Similarly, the tribological properties indicate the material’s possible applications during contact and relative motion with another surface and depend on intrinsic parameters similar to those of adhesion, with the addition of tribo-oxidation, particle formation, and the counterpart’s material [9]. Friction mechanisms are associated with the simultaneous occurrence of adhesion and abrasion processes. The dominance of either of these components depends on the surface characteristics. However, the coefficients of friction and wear undergo changes over time. In the initial stages of motion, friction and wear are primarily influenced by the contact between surface asperities. These asperities undergo cyclic deformation and strain hardening. When this process becomes critical, the asperities fracture, leading to the formation of particles. These particles then undergo a cyclic process of deformation, hardening, and fracturing. As the particles reach a critical size, they are expelled from the wear track, leaving the surface exposed, thus initiating the process anew. However, if the particles possess high hardness and a spherical geometry, they can withstand the contact load and contribute to third-body friction, reducing the coefficient of friction. Conversely, if the particles adhere to the surface, they can give rise to abrasive mechanisms such as scratching and ploughing. In the advanced stages of friction, a steady state is achieved where the coefficient of friction stabilizes. During this stage, the average number of particles formed is equivalent to the number of particles leaving the track, resulting in the surface tending towards a polished state [7,9,10,11]. Titanium nitride (TiN) is a material that can be synthesized using physical vapor deposition (PVD) and is widely used due to its mechanical, tribological properties, and corrosion resistance. However, its usage is limited at high temperatures since it tends to form titanium oxide (TiO_2_) in air at operating temperatures above 600 °C. This phenomenon generates detachment of the film and an increase in wear due to oxidation and tribo-oxidation [3,5,12]. Niobium nitride (NbN) has been used in industry due to its wear resistance, chemical and thermal stability, and high shear strength at temperatures between 800 °C and 1000 °C [3,13,14,15,16,17]. Studies have shown that TiN/NbN multilayer coatings have a hardness range of 23 to 28 GPa, which decreases at temperatures above 800 °C but maintains a higher value compared to their monolayers (TiN and NbN) [18,19,20]. Considering the need to develop coatings with increased wear and corrosion resistance, ternary compounds, such as titanium niobium nitride (TiNbN), combine the properties mentioned above. These coatings provide better adhesion at the film/substrate interface, increased thermal stability and wear resistance, enabling them to reach high operating temperatures [21,22,23].

Deposition of TiNbN has been reported with equiaxial and semi-columnar grains, with a face-centered cubic (FCC) crystalline structure. The atomic radii of Nb and Ti are similar, and thus Nb atoms replace Ti atoms in the TiN structure. However, small differences in radius increase the compressive stresses of the material, enhancing its mechanical properties compared to TiN [24,25]. The available literature indicates that the mechanical properties of the material depend on the chemical composition of the coating, specifically the percentage of Nb, as well as on the film’s morphology, which, in turn, is dependent on the deposition technique. Hardness measurements have shown values ranging between 10 and 26 GPa for values above 25% Nb [25,26,27]. However, there is currently no available information, to the best of our knowledge, on the mechanical properties of TiNbN coatings with low Nb percentages. Although reports on the tribological properties of TiNbN coatings under severe wear conditions are scarce, basic tribological properties have been explored under low load, low speed, and short distance conditions, showing a coefficient of friction of approximately 0.1 and wear rate of approximately 10^−4^ mm^3^/Nm. Abrasion and the formation of hard particles are the main reported wear mechanisms [25,28]. Thus, there is a significant information gap in the information available, and studies are necessary to determine the adhesion, mechanical, and tribological properties of TiNbN coatings under severe wear conditions simulating cutting tool environments, for example. The design of protective coatings offers a means to enhance the longevity of various tool types. However, the existing information regarding the mechanical and tribological behavior of TiNbN coatings fails to demonstrate advancements in their application as surface protection for diverse cutting tools. Considering the material’s favorable mechanical properties and chemical stability, it possesses ideal characteristics for this purpose. Consequently, it is crucial to ascertain the extent of this material’s applicability as a protective coating in simulated cutting environments and to evaluate its tribological properties for suitability. Therefore, this work studied the mechanical properties, adhesion, and tribological properties of TiNbN coatings deposited by cathodic arc, varying the substrate temperature (Ts) to study their resistance in severe wear environments, simulating conditions similar to those of cutting tools.

## 2. Materials and Methods

### 2.1. Sample Preparation

A cylindrical bar, made of AISI D2 steel, with a diameter of 2.5 cm and a length of 100 cm, was cut into cylinders of 7 mm thickness. These cylinders were then subjected to thermal treatment, involving quenching at 1000 °C and followed by cooling in water. They were further subjected to double tempering at 400 °C and cooled in air. Subsequently, the steel surface was prepared according to the ASTM E3-11 standard [29], using a series of silicon carbide (SiC) abrasive papers with increasing grain sizes, ranging from 80 to 2500. Finally, the surface was polished to a mirror finish using 0.1 μm diamond paste.

### 2.2. Coating Deposition

Prior to deposition, the substrates underwent ultrasonic cleaning in alcohol for 15 min, using a 45 kHz sweep frequency in an Elmasonic X-tra US 550 ultrasonic bath (Elma Schmidbauer Gmbh, Singen, Germany). They were then mounted onto a 2-fold rotation substrate holder with a rotation frequency of 2 Hz. TiNbN coatings were then deposited onto silicon (Si) and D2 steel substrates using an Oerlikon Domino Mini Arc-PVD equipment (Oerlikon, Zürich, Switzerland) [9]. Two titanium cathodes with a diameter of 7.62 cm and a purity of 99.995% were used, each with a niobium insert (diameter of 1.27 cm and a purity of 99.99%), which comprised ~1% of the cathode surface. Deposition conditions are presented in Table 1. Coatings were deposited at different substrate temperatures (Ts), specifically 200 °C, 400 °C, and 600 °C, to observe their effects on the adhesion, mechanical, and tribological properties of the surface. The substrates were heated using resistors connected to a 6 kW source, and the temperature was controlled using an Impac 140 infrared pyrometer and three thermocouples located in the sample holder. The deposition process started once the temperature stabilized at the desired value for 60 min. The conditions were optimized to achieve coatings with average thicknesses of 5 μm within a deposition time of ~60 min. Prior to deposition, plasma cleaning was performed using Ar gas for 20 min at a pressure of 0.4 Pa (with a flow rate of 278 sccm) and a pulsed bias voltage of −50 V (at a frequency of 80 kHz with 80% duty cycle).

### 2.3. Coating Characterization

To determine the crystalline structure, X-ray diffraction (XRD) was carried out using a Rigaku SmartLab diffractometer (Rigaku, The Woodlands, TX, USA) with CuKα radiation (λ = 1.5406 nm) at a grazing incidence angle (Ω = 1.5°). The patterns were collected from 20 to 90° in steps of 0.04° for 3 s/step. A Rietveld refinement was then performed for crystallographic analysis to obtain the lattice parameters and crystallite size. For the determination of thickness, chemical composition, and surface morphology, both scanning electron microscopy (SEM) and energy dispersive spectroscopy (EDS) were utilized. These analyses were conducted using a Jeol JSM-7200F FE-SEM microscope (Jeol, Tokyo, Japan) equipped with an Oxford ULTIMAX 100 EDS probe (Oxford Instruments, Abingdon, UK).

To measure roughness, a BRUKER DktakXT 100 contact profilometer (Bruker, Billerica, MA, USA) was employed with a contact load of 3 mg and a 4 mm distance. Eight measurements were taken in different sections of the surface and averaged. Nanoindentation tests were performed using a Hysitron TI 700 UBI equipment (Bruker, Billerica, MA, USA) with a 1 mN load and a 3-faced pyramidal Berkovich indenter. Load–displacement curves were analyzed using the Oliver and Pharr method with a 3 × 3 indentation matrix. Corrections were made for compliance and contact area function using a fused silica reference standard. Vickers hardness was measured in accordance with the ASTM E92-17 standard [30] using a pyramidal-type diamond indenter with a 136° angle between the faces and an applied load of 10 N. Three indentations were made with a dead time of 12 s.

To evaluate the adhesion of the coating to the substrate, scratch tests were conducted in accordance with the ASTM C1624-05 standard [31] using a Rockwell C indenter (200 μm). The tests involved applying a normal load ranging from 0.1 to100 N over a 5 mm distance at a speed of 7 mm/min. SEM images were taken of the areas where cohesive and adhesive failures occurred. Micro-indentation and dynamic scratching were performed using an Anton Paar Revestest (RST3) equipment.

For the determination of the material’s tribological properties, Pin-On-Disk tests were carried out using an Anton Paar THT1000 tribometer, following the ASTM G99-17 standard [32]. The tests were conducted with a 6 mm diameter alumina counterpart (Al_2_O_3_) under a normal load of 5 N, a linear speed of 8.50 cm/s for 5000 cycles (~100 m), and a 3 mm radius. The wear rate (k) was determined using the Archard model (k = V/Ld), where V represents the wear volume obtained by contact profilometry, L is the load, and d is the distance of the tribology test [32].

## 3. Results and Discussion

### 3.1. Crystalline Structure and Chemical Composition

The analysis conducted by XRD (Figure 1a) revealed an FCC crystalline structure with an Fm3m space group. Regardless of Ts, all diffractograms showed a peak shift towards higher diffraction angles compared to the position of the TiNbN phase peaks (black lines on the bottom) (see Figure 1b). This shift is more pronounced at higher angles and can be attributed to a combined effect of Nb inclusion within the TiN lattice (resulting from differences in atomic radii, Ti = 176 pm and Nb = 198 pm) and the generation of internal stress in the thin film due to the increase in Ts. All patterns displayed a monophasic structure (TiNbN) with average calculated lattice parameter (*a*) of 4.31 Å (see Table 2). Since Nb replaces Ti atoms in the TiN crystal lattice, the obtained lattice parameter falls between the values reported for nitrides (TiN = 4.12 A and NbN = 4.37 A) [33,34].

However, the reduction of the lattice parameter with respect to Ts, as observed in Table 2, indicates the formation of compressive stresses in the structure. This phenomenon is associated with the increase in density with higher substrate temperatures [22]. These results also indicate complete solubility between TiN and NbN, which is consistent with findings from other studies in the literature [22,25,28].

The results for crystallite size (Table 2) suggest that the coherent zone is larger, perpendicular (φꞱ) to the incident X-rays compared to parallel (φǁ) orientation. This finding suggest that the crystallite shape is flat-like for low Ts. However, as the temperature increases, the crystals tend to adopt a spherical shape. This transition indicates an improvement in the packing of the crystalline domains within the material as the temperature rises, resulting in smaller crystallographic domains with spherical geometries [22].

The chemical composition of each film is presented in Table 3. An increase in substrate temperature resulted in a decrease in titanium (Ti) content compared to nitrogen (N), while the quantity of niobium (Nb) remained relatively constant. These findings suggest a reduction in the adsorption of Ti atoms due to the elevated substrate temperature [35]. However, the N/Ti ratio increased as the substrate temperature increased, indicating an enhanced reactivity between these two elements [36]. The maximum value of the N/Ti ratio reached 0.99 at a Ts of 600 °C.

### 3.2. Morphology, Thickness, and Roughness

The surface morphology of the films, as depicted in Figure 2, reveals the presence of droplets and pores, which are characteristic features of the cathodic arc deposition process (Figure 2a–c). These surface characteristics significantly influence the mechanical and tribological properties of the surface, as they act as stress concentrators, impacting crack behavior. Cracks can initiate or propagate around sub-micron-sized defects and can be associated with droplets of micrometric sizes [22,36,37,38,39]. The cross-sections of the coatings, illustrated in Figure 2d–f, display a dense columnar growth pattern typical of ceramic coatings such as TiN. However, this growth pattern is interrupted by the presence of molten material droplets that are expelled from the cathode and become embedded in the coating, resulting in the formation of pinholes [22,35,38,40].

Table 4 presents the average roughness (Ra), root mean square roughness (Rq), and the Rq/Ra ratio for the substrate and coatings deposited at different Ts. The surface and cross-sectional profile analysis (see Figure 2) revealed the presence of microdroplets and pores, which contribute to increased roughness values in the coatings compared to the substrate. However, both Ra and Rq decrease with increasing substrate temperature. This can be attributed to the increase in the mobility of the adatoms at higher temperature, leading to the formation of denser coatings, with fewer defects [22]. The Rq/Ra ratio indicates that, regardless of Ts, the roughness distribution tends to be random, with characterized peaks and valleys. This is a result of the initial mechanical polishing performed on the surfaces before deposition and the subsequent formation of droplets and pores.

As mentioned in the experimental details, the process parameters were standardized to achieve an approximate coating thickness of 5 μm. However, it is important to note that the cathodic arc deposition technique may introduce slight deviations from the expected thickness. This variation can be attributed to the random nature of the arc when it interacts with the cathode surface. The arcs move across the surface, generating cathodic spots and causing material evaporation. However, it is not possible to precisely predict the location where the arc will strike, resulting in fluctuations in evaporation and deposition rates. These variations are particularly influenced by the compaction density and surface quality of the cathode. Over time and with an increase in the final thickness of the coating, this phenomenon becomes more noticeable [41,42].

### 3.3. Mechanical Properties

Figure 3a shows the load vs. displacement curves for of the TiNbN coatings for the different Ts. No defects, such as pop-ins, were observed in the curves, indicating the absence of crack formation during the loading stage of the surface deformation process [43,44]. However, increasing the substrate temperature reduces the maximum penetration of the indenter, implying an increase in hardness. Figure 3b shows the hardness (H) values as a function of Ts. A slight increase in H is observed with higher Ts, ranging between ∼29 and ∼32 GPa. This effect is primarily attributed to the increase in compressive stresses within the coatings, which stem from differences in the coefficient of thermal expansion between the film and the substrate, as well as disparities in the atomic radii of Ti and Nb. These consequences were analyzed by XRD, which revealed shifts in the peaks towards higher diffraction angles (Figure 1b). Additionally, an increase in temperature is expected to enhance density [45].

On the contrary, the average value of the elastic modulus (E) (Figure 3b) decreases with increasing temperature, resulting in the increase of the coefficients H/E (elastic strain limit for failure) and H^3^/E^2^ (plastic deformation resistance) with Ts (Figure 3c). However, the statistical distribution of data obtained for E indicates that the coatings exhibit similar values for different Ts. Consequently, when considering the propagation of statistical deviation, the value of H/E does not show a significant change as a function of Ts. These results suggest that more energy is required for the generation of sudden cracks, and the initial contact tends to be more elastic [43,44]. As observed in the chemical composition analysis, increase in the atomic percentage of nitrogen (see Table 3) and hardness were observed with increasing temperature. This is attributed to the greater number of covalent bonds, which are stronger than metallic bonds [46]. Furthermore, the average increase in H/E and H^3^/E^2^ implies a more rigid crystal structure upon contact. A similar phenomenon has been observed in other transition metal nitrides [27,47].

### 3.4. Adhesion

The adhesion of the coatings was determined by dynamic scratch tests. Table 5 shows the critical loads of cohesion (Lc1) and adhesion (Lc2), calculated from the drag coefficient, acoustic emission, and load vs. distance plots. These values were averaged from three separate measurements. Figure 4 shows an overview of the scratch tracks and SEM images of the sections from the scratch track where critical loads were obtained. A decrease in Lcs was observed with increasing substrate temperature. As discussed in the structural, morphological, and mechanical properties sections, the increase in compressive stresses with Ts also leads to an increase in the rigidity of the coatings. This rigidity enhances the generation of cracks and the detachment of coating material with high stresses, resulting in decreased adhesion, despite the improvement in the mechanical properties of the system [27,45]. The observed increase in hardness (Figure 3b) does not guarantee improve adhesion. A harder coating is generally less prone to scratching or removal from the substrate, indicating better adhesion. However, the slight reduction in elastic modulus (Figure 3b) and the increase in plastic deformation resistance (Figure 3c) generally lead to a reduction in the resistance to crack initiation and propagation at the film/substrate interface [27,35,44]. This indicates that the stresses generated at the substrate/film interface play a crucial role in coating design and that increasing the mechanical properties of the surface must be balanced with the formation of coherent interfaces during deposition.

However, on demonstrating the repeatability of the scratch test, similar failure types were observed regardless of the substrate temperature. Figure 4 shows an overview of the scratch tracks and SEM images of the cohesive (Lc1) and adhesive (Lc2) failures for the different substrate temperatures. The cohesive failures observed correspond to lateral cracks, which are micro-cracks occurring at the edges of the track that opens parallel to the scratch direction. Additionally, the coating deposited at Ts = 400 °C shows conformal cracks, forming arcs that open away from the direction of scratching. These cracks generates because the film tries to conform to the plastic deformation of the substrate. On the other hand, adhesive failures were related to chipping spallation, resulting in rounded removal regions of coating extending laterally from the edges of the track. The film deposited at Ts = 600 °C displays adhesive cracks that form perpendicular to the scratch direction, originating from the adhesive failures (chippings). These cracks develop at the interface between the film and the substrate and are more common in brittle coatings with high thicknesses. This observations aligns with the analysis conducted on the mechanical properties, where this particular coating exhibits the highest hardness and the lowest elastic modulus among the entire set.

The dynamic scratch analysis performed showed that at lower substrate temperatures, higher adherence was observed, accompanied by less severe failures, as observed in the case of Ts = 200 °C. In contrast, higher substrate temperatures resulted in cohesive and adhesive failures occurring at lower loads. This behavior is attributed to the lower plastic deformation resistance of the film (Figure 3c), allowing it to accommodate the substrate deformation and leading to adhesive failure at higher loads.

Furthermore, an increase in substrate temperature enhances the mobility of species within the substrate. This increased mobility leads to stabilization in higher energy sites, which can either improve or worsen the adhesion forces. Therefore, the temperature increase is associated with higher energy accumulation at the interface, facilitating the thermal diffusion of the coating towards the substrate. Additionally, elevated substrate temperatures contribute to the generation of compressive stresses, which can have a negative impact on adhesion [22,48,49].

### 3.5. Cracking Patterns

Figure 5 presents the images of the Vickers indentations created on the coated surface. The presence of roughness, pores, and microdroplets (as shown in Figure 2), along with variations in compressive stresses throughout the thickness and the accumulation of elastic energy in the coating due to increasing substrate temperature, contribute to induced failures in the material. These failures manifest as cracks parallel to the edges of the indentation (referred to as picture frame cracking) and partial delamination of the coating. Picture frame cracking occurs due to the plastic deformation of the steel, which causes the coating to bend [50,51] These failures may also originate from pores or pinholes formed during the deposition process.

Considering the mechanical properties of the surfaces, where the hardness and plastic deformation resistance of the film exceed those of steel, the formation of picture frame cracking on the deformed surface is expected, as previously observed in TiN coatings [50]. Additionally, the variation in substrate temperature leads to the release of elastic energy stored within the coatings, resulting in cracks propagating across the interface and ultimately causing coating detachment [52].

Edge cracks were observed in the coatings deposited at substrate temperatures of 400 °C and 600 °C (Figure 5b,c respectively). This type of failure occurs when there is an increase in compressive stresses, allowing the crack to propagate through the interface between the substrate and the coating and extend beyond the indentation. The high elastic deformation and elastic recovery produced in the substrate, also contribute to the formation of cracks that follows a circular path in external areas to the indentation [50,52]. As indicated in Table 3, these coatings have higher thickness, which suggests an increase of compressive stress. Moreover, the observation of peak shifting in the diffraction patterns (Figure 1b) indicates an increase in the compressive stresses of the coatings with the increase in Ts. The formation of edge cracking is also associated with the decrease in elastic moduli and the increase in H/E and H^3^/E^2^ ratios [26], as observed in the nanoindentation results (Figure 3). The appearance of edge cracks is further influenced by the reduction in adhesion observed in Table 5 and Figure 4.

### 3.6. Wear Resistance

Figure 6 shows the coefficient of friction (COF) vs. cycles for D2 steel and the coatings deposited at different Ts, with values in the stabilization stage of 0.95 (substrate), 0.45 (Ts = 200 °C), 0.55 (Ts = 200 °C), and 0.8 (Ts = 200 °C). The coatings showed a lower COF compared to the D2 steel; however, the friction processes differ, particularly in the early stages of the test. For the coatings, during the first 1000 cycles, there is a sudden increase in the COF, indicating a rapid breakdown of asperities. This phenomenon is attributed to the deformation and fracture of microdroplets formed during the cathodic arc deposition process, which exhibit lower energy thresholds for deformation and breakage (see Figure 2). In contrast, the D2 steel undergoes a prolonged stage of asperity breaking, as the contact surfaces between the steel and the pin have lower roughness and deformability, leading to early contact between polished or smooth surfaces. This observation aligns with the roughness values presented in Table 4.

In all materials, the stabilization stage is observed from ~1000 cycles, during which more energy is required for particle breaking. These particles, depending on the material, tend to be harder and more brittle as a result of hardening through plastic deformation. For instance, in the case of the coatings deposited at Ts = 600 °C (with a hardness of ~32 GPa) and Ts = 400 °C (with a hardness of ~30 GPa), the abrasive component of friction is expected to dominate and its stabilization stage exhibits greater instability compared to the coating deposited at lower substrate temperatures. This suggests higher particle generation or wear debris processes. The coatings deposited at Ts = 200 °C exhibit a lower COF, which can be attributed to their lower hardness (around 28 GPa) and lower resistance to plastic deformation. However, the improved mechanical properties of these surfaces compared to D2 steel enable them to withstand the applied loads, resulting in a reduced COF.

In Figure 7, SEM images of the wear tracks for the different materials are observed, along with an EDS elemental chemical mapping insert for oxygen. Figure 7a shows the wear track of the D2 steel substrate, where adhesion and ploughing are the dominant wear mechanisms. Highly deformed particles are observed in the center of the track, accompanied by a high concentration of oxygen throughout the wear track. Figure 7b shows the wear track of Ts = 200 °C, revealing signs of micro-scratching and the presence of particles with high deformation in the center of the track. As discussed from the mechanical properties (see Figure 3), this film deposited at the lower temperature showed the lowest hardness and plastic deformation resistance (H^3^/E^2^), indicating that the debris can deform with lower energy, resulting in reduced abrasiveness during the wear test. This cyclic process of particle deformation, hardening, and breaking, which occurs at a slower rate for this deposition condition, leads to a reduction of the COF as observed in Figure 6. Figure 7c shows the wear track of Ts = 400 °C. Similar to the observations at lower substrate temperatures, deformed particles and micro-scratch grooves are formed in the center of the track. Adhered particles can be seen at the edges of the track, and together with the EDS oxygen analysis, it can be inferred that they consist of oxides of the coating material, as no presence of Fe was observed on the surface. These wear particles, expelled from the track, showed high oxygen content and contribute to increased wear and abrasion on the contact surfaces, which explains the observed increase in COF in Figure 6. Figure 7d shows the wear track of Ts = 600 °C. An increase in abrasion, as well as in the amount of oxygen in the worn surface and the presence of particles in the center and in the corners of the wear track are observed. Despite the detected increase in hardness and plastic deformation resistance for this substrate temperature, the formation of particles intensifies the abrasive component of friction, resulting in an elevated COF (see Figure 6).

The observed wear mechanisms are consistent with the wear rate (k) depicted in Figure 8. The D2 steel substrate shows a high value of k, which is attributed to the ploughing generated by numerous particles adhered to the surface. In contrast, the coatings deposited at different temperatures demonstrate lower wear rates (approximately one order of magnitude lower). However, in alignment with the hardness trend, the wear rate increases with the substrate temperature. This increase in wear rate is associated with the increase in abrasion resulting from the presence of hard particles. These particles undergo a cyclic process of deformation, hardening, and breaking. Eventually, they are expelled from the wear track and accumulate at the edges, exposing the bare surface for the repetition of the wear process. This phenomenon was analyzed in the coatings deposited at Ts of 400 °C and 600 °C (see Figure 7).

In the case of the substrate temperature of Ts = 200 °C, the aforementioned process took a longer time, and a significant number of particles were observed within the wear track. Although the friction coefficients obtained in this study are higher than those reported in the literature, the wear rate values are approximately one order of magnitude lower [25]. This indicates an improvement in the wear resistance of the material compared to results obtained by other researchers under conditions of severe wear, higher loads, and longer distances.

The tribological behavior of coated surfaces is influenced by various intrinsic and extrinsic factors. Intrinsic factors include parameters associated with film deposition and the counterpair, such as roughness, chemical composition, the relationship between the mechanical properties of the film, substrate, and the counterpair, as well as the stress state at the interface, which directly affects the adhesion of the coating. On the other hand, extrinsic factors include load (and consequently contact stress), speed, type of movement, atmospheric conditions, temperature, and time (or distance or number of cycles) [9,53]. The complexity and dynamic nature of these factors often lead to contradictory trends when evaluating the effect of substrate temperature on the tribological properties of coatings, as they directly influence the intrinsic parameters.

For instance, studies conducted on TiN and WTiN coatings deposited by DC magnetron sputtering have shown that increasing the substrate temperature leads to an increase in both the coefficient of friction (COF) and the wear rate (k). In the case of TiN coatings, this increase is attributed to a reduction in hardness, an increase in the modulus of elasticity, and a decrease in the H/E ratio with increasing Ts. However, no data regarding coating adhesion were reported [54]. On the other hand, for WTiN coatings, the increase in COF is attributed to a reduction in the critical load Lc1 with increasing Ts, but no data on mechanical properties or critical loads Lc2 were provided [55]. It should be noted that the counterparts and contact stresses were similar in both studies. In comparison to these previous research works, the present study observed an increase in hardness, a slight reduction in the average value of E (elastic modulus), and an increase in the average value of H/E (hardness-to-modulus ratio), resulting in a decrease in adhesion.

Furthermore, studies on TaN coatings deposited by magnetron sputtering have reported that increasing Ts leads to a reduction in COF. In this case, the improvement in tribological properties is attributed solely to the change in the crystallographic texture of the material with substrate temperature, without providing data on mechanical properties or material adhesion [56]. On the other hand, coatings such as WC and AlCrWTiMoN did not exhibit changes in COF as a function of increasing substrate temperature, despite reporting an increase in mechanical properties [57,58].

Numerous researchers have investigated the tribological properties of TiN coatings under conditions simulating severe wear encountered in cutting tools. These studies have shown that the friction coefficients of TiN can range from 0.3 to 0.8, depending on the specific contact conditions. Under high loads and moderate speeds (<50 cm/s), the coefficient of friction tends to decrease due to the formation of lubricating oxide layers. However, wear rates significantly increase, exceeding 30 × 10^−4^ mm^3^/Nm [59,60,61,62]. It has also been observed that dominant wear mechanisms for TiN coatings include abrasion, scratching, and ploughing caused by hard particles. However, most references primarily focus on coatings with low thicknesses (<2 μm), resulting in relatively short lifetimes of less than 2000 cycles. After wear occurs, even partially, both the coefficient of friction and wear rate tend to increase, reaching values as high as 0.9 and 40 × 10^−4^ mm^3^/Nm, respectively [59,60,62]. Similarly, studies have demonstrated that incorporating elements that stabilize oxide formation, such as Al in TixAlyN [62,63], Hf in TixHfyN [9], and Nb in TiNbN [25], can lead to reductions in the coefficient of friction and abrasion at both low and high temperatures. These modifications have resulted in an extended useful life for the surface, ultimately enhancing the overall performance of the material in applications including cutting tools, cold and hot forming and drawing, erosion, and impact [63,64].

Therefore, it is necessary to analyze the influence of intrinsic parameters on the wear resistance of coated surfaces on a case-by-case basis. In the case of TiNbN coatings deposited in this study, in contact with Al_2_O_3_ balls, the tribological response is associated with the increase in mechanical properties and the reduction in adhesion with Ts, leading to the presence of abrasive particles that increase the COF and wear. However, the deposition of these coatings reduces wear on the substrate by an order of magnitude.

## 4. Conclusions

The tribological properties of TiNbN coatings, deposited using cathodic arc, were investigated under severe wear conditions that simulated cutting environments. The temperature of the substrate was varied during the study. The coatings exhibited friction coefficients ranging from 0.45 to 0.8, and wear rates between 0.08 and 0.23 × 10^−4^ mm^3^/Nm were observed. These values were lower than those observed for the AISI D2 steel substrate. Additionally, a quasi-stable friction stage was observed after approximately 1000 cycles. The wear processes observed on the surface of the TiNbN coatings were primarily associated with abrasion mechanisms, particularly involving third-body interactions (scratching and ploughing) and tribo-oxidation. These mechanisms are highly beneficial for cutting tools, as the formation of hard particles enhances material removal capacity, while the formation of stable oxides improves the thermal stability of the tool.

The behavior of the wear particles dominated the tribological response of the surfaces. The surface characteristics were especially influential in the TiNbN coatings deposited by cathodic arc, due to the formation of microparticles and pores, typical of the coatings obtained by this technique.

For the D2 steel substrate, particles adhering to the surface and ploughing were observed, which generated a high friction coefficient and wear. When depositing the TiNbN coatings with different temperatures, a reduction of the wear rate of one order of magnitude was observed. However, by increasing the substrate temperature, particles with brittle behavior and smaller size increased abrasion and wear. This was related to the increase in hardness and plastic deformation resistance and a decrease in adhesion with substrate temperature.

Through EDS elemental chemical mapping, an increase in the oxygen during wear was observed with increasing substrate temperature. While in the D2 steel substrate, oxygen is concentrated inside the wear track, in the coatings an increase was observed at the edges, which was accentuated with the increase in substrate temperature. This is associated with the presence of chemically activated particles, which showed plastic deformation and rupture.

An increase in the substrate temperature reduces both the cohesive load (Lc1) and the adhesive load (Lc2). However, the failure mechanisms found were similar, regardless of temperature, and were related to lateral and conformal cracks for Lc1 and chipping for Lc2.

A similar effect was observed in the cracking patterns generated by micro-indentation, where picture frame cracks were generated within the indentation regardless of the substrate temperature. However, when increasing the temperature, the generation of radial cracks was observed, which propagate through the interface between the substrate and the coating. These types of cracks are associated with deformation of the substrate away from the indentation and the inability of the coating to “accommodate” the plastic conditions and a reduction in adhesion is indicated.

## Figures and Tables

**Figure 1 materials-16-04531-f001:**
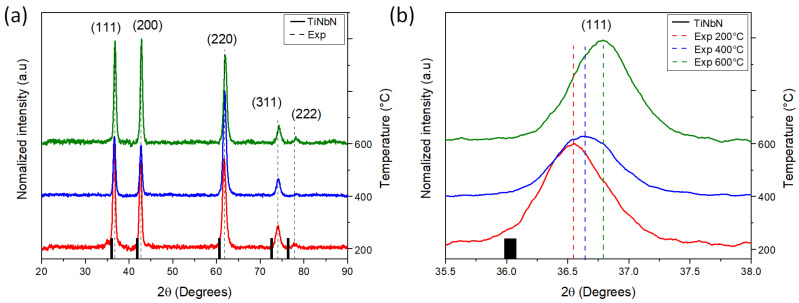
(**a**) X-ray diffraction of coatings deposited at Ts 200 (red), 400 (Blue), and 600 °C (green), (**b**) zoom of the peaks (111) for the different temperatures.

**Figure 2 materials-16-04531-f002:**
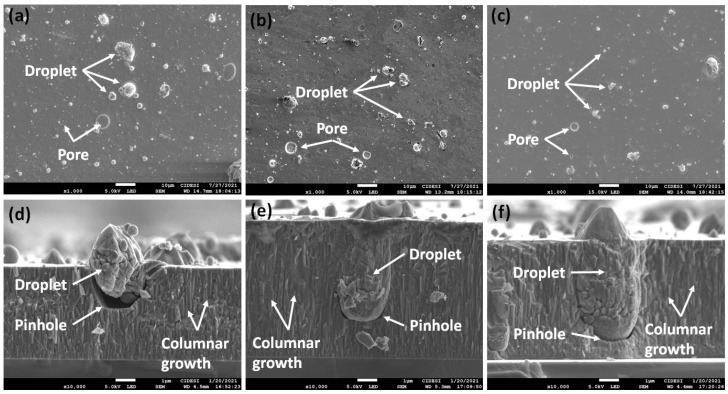
SEM images of the coating surface deposited at different substrate temperatures Ts: (**a**) 200 °C, (**b**) 400 °C, (**c**) 600 °C and cross-section of the coatings deposited using (**d**) 200 °C, (**e**) 400 °C, and (**f**) 600 °C.

**Figure 3 materials-16-04531-f003:**
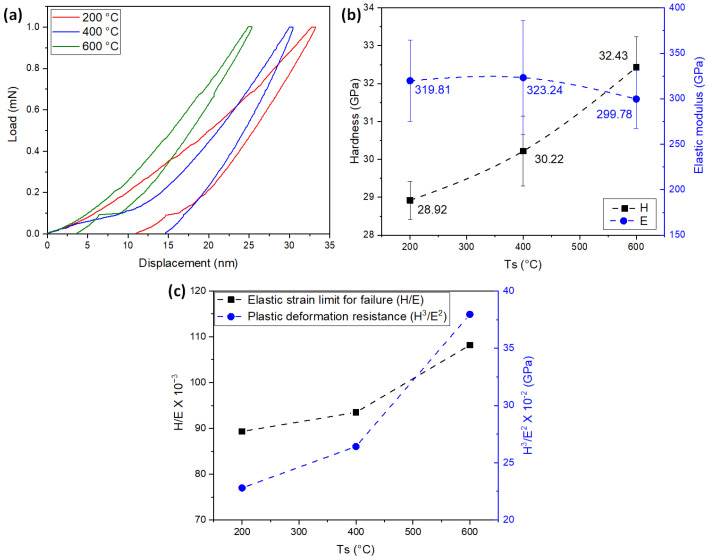
Mechanical properties obtained by nanoindentation. (**a**) Representative load–displacement plot for different Ts, (**b**) hardness and elastic modulus, and (**c**) elastic strain limit for failure (H/E) and plastic deformation resistance (H^3^/E^2^).

**Figure 4 materials-16-04531-f004:**
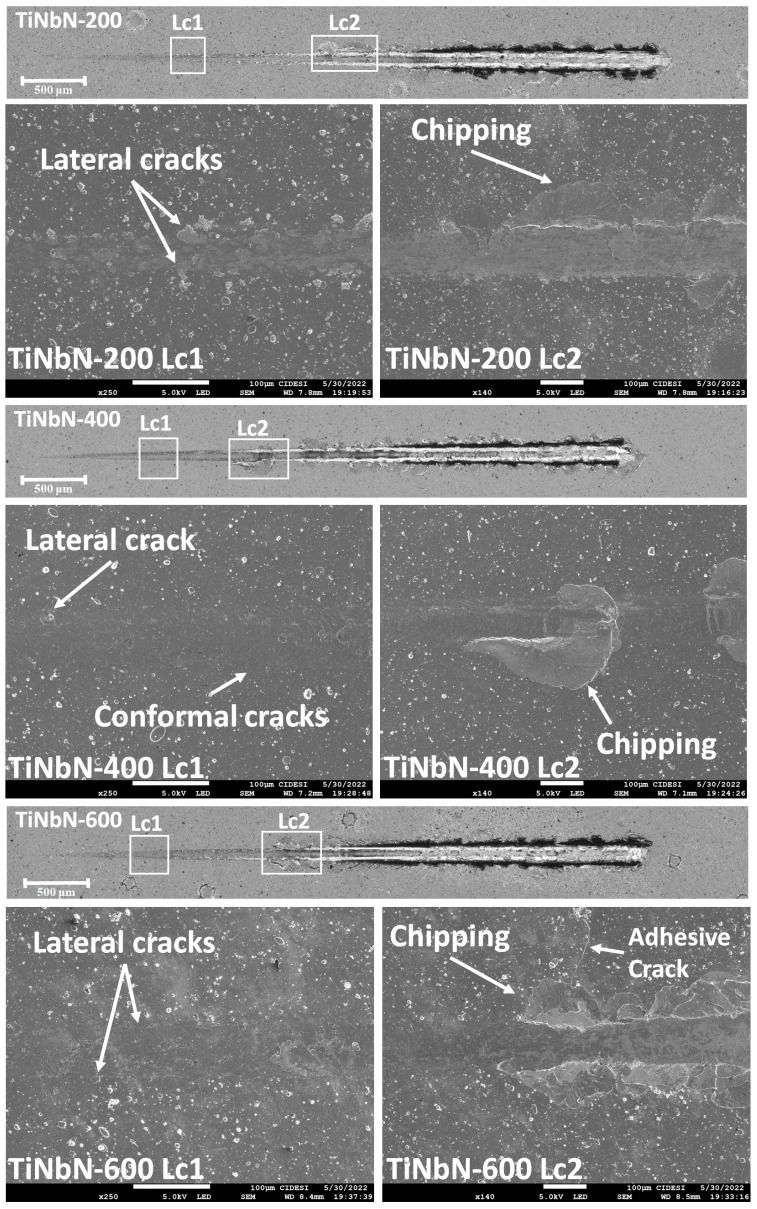
Overview of the scratch tracks and SEM images of the failures on the critical load obtained from the scratch test for the coatings deposited at different Ts.

**Figure 5 materials-16-04531-f005:**
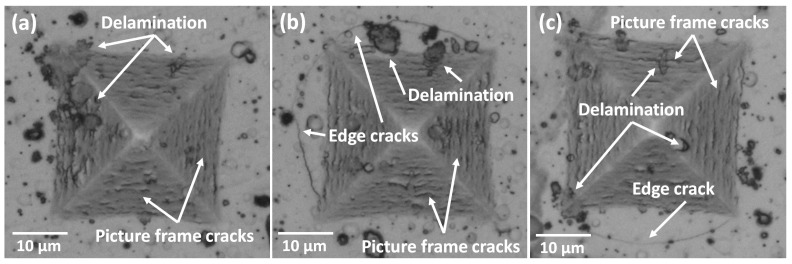
Indentation images at 10 N of TiNbN coating deposited at (**a**) Ts = 200 °C, (**b**) Ts = 400 °C, and (**c**) Ts = 600 °C.

**Figure 6 materials-16-04531-f006:**
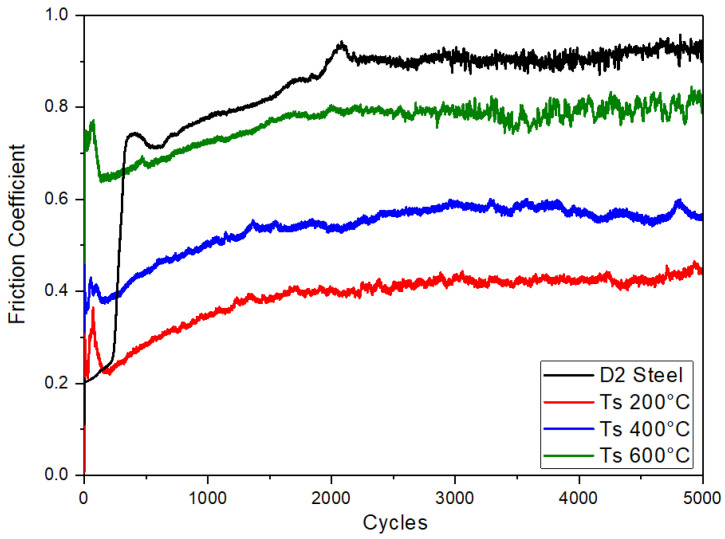
Coefficient of friction vs. cycles for the D2 steel substrate and the films deposited at different temperatures.

**Figure 7 materials-16-04531-f007:**
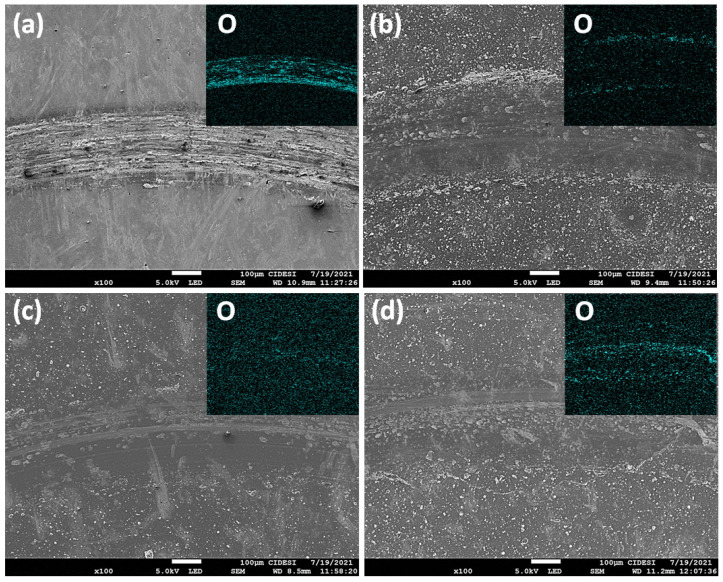
SEM images of the wear track, including the oxygen (O) EDS mapping insert for: (**a**) AISI D2 steel, (**b**) Ts = 200 °C, (**c**) Ts = 400 °C, and (**d**) Ts = 600 °C.

**Figure 8 materials-16-04531-f008:**
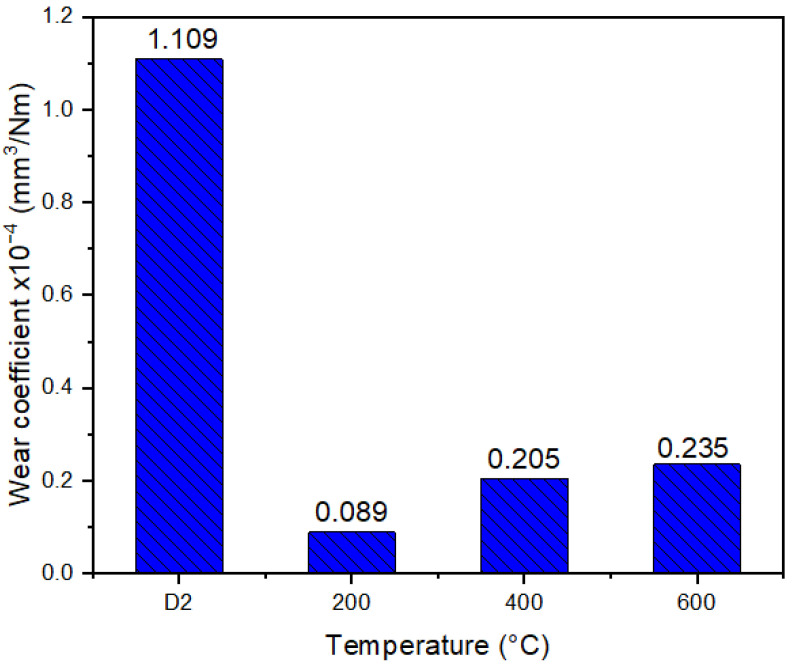
Wear rate for the D2 steel substrate and the coatings deposited at different temperatures.

**Table 1 materials-16-04531-t001:** Coating deposition conditions.

Pressure (Pa)	Nitrogen Flow (sccm)	Power (kW)	Bias Voltage (V)	Temperature (°C)
0.4	500	3	−100	200
				400
				600

**Table 2 materials-16-04531-t002:** Structural and composition results of coatings of TiNbN at different temperatures; *a* indicates the lattice parameter, φꞱ perpendicular and φǁ parallel crystalline size.

Ts (°C)	*a* (±0.02 Å)	φꞱ (±0.1 nm)	φǁ (±0.1 nm)
200	4.35	72.1	11.0
400	4.34	52.5	18.3
600	4.24	41.4	27.3

**Table 3 materials-16-04531-t003:** Chemical composition and coating thickness.

Ts (°C)	Ti (at%)	Nb (at%)	N (at%)	Thickness (µm)
200	57.86 ± 7.28	0.21 ± 0.01	41.93 ± 7.27	4.68 ± 0.03
400	51.06 ± 0.57	0.18 ± 0.02	48.76 ± 0.58	6.80 ± 0.12
600	44.72 ± 3.09	0.15 ± 0.02	55.12 ± 3.08	5.93 ± 0.04

**Table 4 materials-16-04531-t004:** Roughness of the D2 steel and TiNbN coatings.

Ts (°C)	Ra (nm)	Rq (nm)	Rq/Ra
D2 Steel	64.27 ± 24.05	78.67 ± 26.44	1.24
200	358.70 ± 261.26	702.12 ± 417.52	2.04
400	141.27 ± 38.66	263.91 ± 85.39	1.85
600	179.33 ± 33.96	340.78 ± 77.81	1.89

**Table 5 materials-16-04531-t005:** Critical load obtained from the scratch test.

Ts (°C)	Lc1 (N)	Lc2 (N)
200	36.9 ± 0.18	53.63 ± 0.27
400	22.55 ± 0.11	48.57 ± 0.24
600	18.45 ± 0.09	38.45 ± 0.19

## Data Availability

The data presented in this study are available on request from the corresponding author. The data are not publicly available because the results correspond to partial data from a broader investigation financed by CONAHCYT and the University of Ibagué and the information associated with future technological developments will be subject to intellectual property protection.

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
