# Peer review of "TiNbN Hard Coating Deposited at Varied Substrate Temperature by Cathodic Arc: Tribological Performance under Simulated Cutting Conditions"

_materials, 2023, doi:10.3390/ma16134531_

Round 1

Reviewer 1 Report

Manuscript ID: Materials-2446113-V1

The manuscript titled "Effect of substrate temperature on adhesion and tribological properties of TiNbN coatings deposited by cathodic arc" investigates the impact of varying substrate temperature (200, 400, and 600 ºC) on the mechanical and tribological properties of TiNbN films. The objective of this study is to determine the optimal deposition temperature that renders TiNbN films suitable for cutting tool applications, specifically in harsh environments.

The following suggestions may improve the proposed manuscript:

Main comments:

1.     The title focuses on the effects of varying Ts on film properties, while the research problem pertains to the lack of information regarding the testing of mechanical and tribological properties in severe environments that simulate real cutting conditions. Could you change the title to agree with the research aim that answers the research question or vice versa?

2.     In the introduction section, the authors should explain wear and friction mechanisms, highlighting the sources of particle formation and their impact on wear and friction behavior.

3.     The research problem is not clear. Why this study is important?

4.     The testing parameters of adhesion, wear, and friction simulate real cutting conditions should be discussed comparing with those in literatures. Addressing this question will enhance the novelty of the research work.

5.     Why films hardness increases while the elastic modulus decreases? Generally, the hardness is expected to be around 10% of the elastic modulus. Clarifying this discrepancy will provide a better understanding of the observed results.

6.     The impact of varying film thickness on properties such as hardness, elastic modulus, adhesion, toughness, wear resistance, and coefficient of friction should be addressed. It is important to discuss how changes in film thickness may affect these properties because the thickness should be fixed for all the studied films.

Introduction section

7.     (Page no. 2, lines 78-80) the statement is difficult to understand.

8.     (Page no. 2, lines 80-81) write hardness measurements rather than nanoindentation data because nanoindentation data include hardness and elastic modulus.

9.     (Page no. 2, lines 85-86) write wear rate instead of wear coefficient and coefficient of friction instead of friction coefficient.

Materials and Methods:

10.  Describe in which bath you did substrates-cleaning via ultrasonic?

11.  (Page no. 3, lines 116) Do you mean duty cycle instead of “live time”?

Results section:

12.  Could you use chemical composition instead of “elemental chemical composition” through the manuscript?

13.  Could you use compressive stress instead of “residual stress” through the manuscript?

14.  (Page no. 4, line 159) Why the authors claim that increasing the Ts leads to increase the compressive stress while there is no increase in the shifting-value of XRD peaks with increasing Ts?

15.  Regarding hardness measurements, it is recommended to employ a load in the range of 10-20 mN. Additionally, it is important to ensure that the indentation depth is less than or close to 10% of the film thickness to obtain accurate hardness values.

16.  The adhesion measurement can be influenced by several factors, including the hardness, toughness, and thickness of the films. These effects should be discussed in the manuscript to provide a comprehensive analysis of the adhesion properties.

17.  In Figure 4, it would be helpful to zoom out the image of lateral and conformal cracks to improve their visibility and distinguishability. This would enhance the clarity of the results and facilitate a better understanding of the crack morphology.

18.  Is it correct to describe the pinhole/droplets sites as delamination in Fig.5?

Conclusion section:

19.  The conclusion should be modified to clearly address the research question and highlight the novelty and significance of the study.

The authors are encouraged to enhance the clarity and readability of the English language used in the manuscript.

Author Response

Jeferson Piamba Jimenez

Professor

Universidad de Ibague

Carrera 22 Calle 67 B, Av. Ambala, Ibagué.

Tolima, Colombia

[email protected]

June 13, 2023

Response to Reviewer 1

Dear Reviewer 1,

First I want to thank you for taking the time to review the paper titled "EFFECT OF SUBSTRATE TEMPERATURE ON ADHESION AND TRIBOLOGICAL PROPERTIES OF TiNbN COATINGS DEPOSITED BY CATHODIC ARC" and making your constructive and highly relevant comments. All the reviews were answered and we put them to your consideration for the publication of the paper.

Below I explicitly describe each of the changes based on feedback:

The manuscript titled "Effect of substrate temperature on adhesion and tribological properties of TiNbN coatings deposited by cathodic arc" investigates the impact of varying substrate temperature (200, 400, and 600 ºC) on the mechanical and tribological properties of TiNbN films. The objective of this study is to determine the optimal deposition temperature that renders TiNbN films suitable for cutting tool applications, specifically in harsh environments.

The following suggestions may improve the proposed manuscript:

Main comments:

  1. The title focuses on the effects of varying Ts on film properties, while the research problem pertains to the lack of information regarding the testing of mechanical and tribological properties in severe environments that simulate real cutting conditions. Could you change the title to agree with the research aim that answers the research question or vice versa?
  • ANSWER// We appreciate the reviewer's comment. In this sense we agree and we decided to modify the title of the paper, leaving:

"Tribological properties under simulated cutting conditions of TiNbN coatings deposited by cathodic arc varying the substrate temperature"

  1. In the introduction section, the authors should explain wear and friction mechanisms, highlighting the sources of particle formation and their impact on wear and friction behavior.
  • ANSWER// We appreciate the reviewer's comment and consider it very convenient. In order to improve the context in relation to friction and wear, a paragraph was included in the introduction, where it is elaborated on the mechanisms of friction and wear with emphasis on the mechanisms of particle formation and its influence on friction and wear, including new references on the subject:

“Friction mechanisms are associated with the simultaneous occurrence of adhesion and abrasion processes. The dominance of either component depends on the surface characteristics. However, the coefficient of friction and wear undergo changes over time. In the initial stages of motion, friction and wear are primarily influenced by the contact between surface asperities. These asperities undergo cyclic deformation and strain hardening. When this process becomes critical, the asperities fracture, leading to the formation of particles. These particles then undergo a cyclic process of deformation, hardening, and fracturing. As the particles reach a critical size, they are expelled from the wear track, leaving the surface exposed, thus initiating the process anew. However, if the particles possess high hardness and a spherical geometry, they can withstand the contact load and contribute to third-body friction, reducing the coefficient of friction. Conversely, if the particles adhere to the surface, they can give rise to abrasive mechanisms such as scratching and ploughing. In the advanced stages of friction, a steady state is achieved where the coefficient of friction stabilizes. During this stage, the average number of particles formed is equivalent to the number of particles leaving the track, resulting in the surface tending towards a polished state [7, 9-11].

  1. The research problem is not clear. Why this study is important?
  • ANSWER// we consider the reviewer's comment important. In order to broaden the discussion around the research problem, the introduction was expanded, including the paragraph:

“The design of protective coatings offers a means to enhance the longevity of various tool types. However, the existing information regarding the mechanical and tribological behavior of TiNbN coatings fails to demonstrate advancements in their application as surface protection for diverse cutting tools. Considering the material's favorable mechanical properties and chemical stability, it possesses ideal characteristics for this purpose. Consequently, it is crucial to ascertain the extent of this material's applicability as a protective coating in simulated cutting environments and to evaluate its tribological properties for suitability.

Additionally, we consider that the last paragraph of the introduction includes the necessary data for the reader to understand the motivation of the work:

“Therefore, this work studies the mechanical properties, adhesion, and tribological properties of TiNbN coatings deposited by cathodic arc, varying the deposition substrate temperature (Ts) to study their resistance in severe wear environments, simulating conditions similar to those of cutting tools.”

  1. The testing parameters of adhesion, wear, and friction simulate real cutting conditions should be discussed comparing with those in literatures. Addressing this question will enhance the novelty of the research work.
  • ANSWER// We appreciate the reviewer's valuable comment, and we recognize the significance of expanding the discussion on this topic. To address this, we have included additional paragraphs in the Results and Discussion section, which specifically focus on the influence of process parameters in relation to the approximation of cutting conditions and their impact on the tribological properties:

“Various researchers have conducted investigations on the tribological properties of TiN under conditions that simulate severe wear scenarios resembling those encountered in cutting tools. These studies have revealed that the friction coefficients of TiN can range between 0.3 and 0.8, depending on the specific contact conditions. When subjected to high loads and moderate speeds (<50 cm/sec), the coefficient of friction tends to decrease due to the formation of lubricating oxide layers. However, wear rates increase significantly, surpassing 30 x 10-4 mm3/Nm [56-59]. Furthermore, it has been observed that the dominant wear mechanisms for TiN coatings include abrasion, scratching, and ploughing caused by hard particles. Nevertheless, most of the references available primarily focus on coatings with low thicknesses (<2 μm), resulting in relatively short lifetimes of less than 2000 cycles. After the coating undergoes wear, which may be partial, both the coefficient of friction and the wear rate tend to rise, reaching values as high as 0.9 and 40 x 10-4 mm3/Nm, respectively [56, 57, 59]. Similar-ly, studies have demonstrated that incorporating elements that stabilize oxide formation, such as Al in TixAlyN [59, 60], Hf in TixHfyN [9], and Nb in TiNbN [25], can lead to reductions in the coefficient of friction and abrasion at both low and high temperatures. These modifications have resulted in an extended useful life for the surface, ultimately enhancing the material's overall performance in applications that include cutting tools, cold and hot forming and drawing, erosion and impact [60, 61].”

  1. Why films hardness increases while the elastic modulus decreases? Generally, the hardness is expected to be around 10% of the elastic modulus. Clarifying this discrepancy will provide a better understanding of the observed results.
  • ANSWER// we appreciate the reviewer's comment and consider it appropriate. However, As analyzed in figure 3b, a slight increase in the hardness of the material is observed, and only a small reduction in the average value of the elastic moduli was observed, but the statistical distribution of the data (the standard deviation of results represented by the bars in each data) indicate that there is no real statistical difference between the values obtained for the three substrate temperatures. Taking into account the reviewer's comment, with which we agree, and taking into account these deviation values, it is clearly true that the hardness is approximately 10% of the value of the modulus of elasticity.

This fact was located in the analysis of mechanical properties, in the paragraph:

“On the contrary, the average value of the Elastic modulus (E) (Figure 3b) is reduced with the increase in temperature and therefore the H/E (elastic strain limit for failure) and H3/E2 (plastic deformation resistance) coefficients increase with Ts (Figure 3c). However, the statistical distribution of data obtained for E indicates that the coatings showed similar values for different Ts and for this reason, when propagating the statistical deviation, the value of H/E did not show a significant change as a function of Ts.”

  1. The impact of varying film thickness on properties such as hardness, elastic modulus, adhesion, toughness, wear resistance, and coefficient of friction should be addressed. It is important to discuss how changes in film thickness may affect these properties because the thickness should be fixed for all the studied films.
  • ANSWER// The reviewer's comment is important. In order to improve the analysis on the subject, the following paragraphs were included:

“As indicated in the experimental details, the process parameters were standard-ized to achieve an approximate coating thickness of 5 μm. However, it should be noted that the cathodic arc deposition technique may result in slight deviations from the ex-pected thickness. This is attributed to the random nature of the arc when it interacts with the cathode surface. The arcs move across the surface, generating cathodic spots and causing material evaporation. However, it is not possible to predict the precise lo-cation where the arc will strike, and as a result, variations in evaporation and deposi-tion rates can occur, particularly influenced by the compaction density and surface quality of the cathode. This phenomenon becomes more pronounced over time and with an increase in the final thickness of the coating. [38, 39]”.

In terms of mechanical properties, the nanoindentation tests were performed using a low load, so that the depth of penetration is always less than 10% of the total thickness of the coating, which implies that the results obtained are not influenced by the thickness of the coating.

Regarding adhesion, the tests were intentionally designed with a low advance speed (7 mm/min) to ensure that it does not significantly impact the cohesive critical load values, even in the presence of variations in coating thickness. Additionally, as depicted in Figure 4, it can be observed that the cohesive failure type (Lc1) remains consistent across different substrate temperatures, regardless of the observed thickness differences. This indicates that, despite these variations, similar crack formation mechanisms occur, primarily attributed to compression failure resulting from the movement of the indenter across the surface. Consequently, no changes in the adhesion states were observed due to the thickness of the coating.

Regarding wear resistance, variations could be observed depending on the thickness, including premature wear and increased abrasion due to the emergence of substrate particles. However, Figure 7 demonstrates that no anomalous events were observed based on the observed thickness differences. On the contrary, the observed behavior can be attributed to mechanisms associated with the mechanical properties, roughness, and temperature of the substrate, which are not influenced by the coating thickness.

Introduction section

  1. (Page no. 2, lines 78-80) the statement is difficult to understand.
  • ANSWER// We acknowledge the reviewer for his comment. The sentence was modified by: “The available literature indicates that the mechanical properties of the material depend on the chemical composition of the coating, specifically the percentage of Nb, as well as on the film's morphology, which, in turn, is dependent on the deposition technique.”
  1. (Page no. 2, lines 80-81) write hardness measurements rather than nanoindentation data because nanoindentation data include hardness and elastic modulus.
  • ANSWER// We consider the reviewer's comment to be correct and the corresponding modification was made
  1. (Page no. 2, lines 85-86) write wear rate instead of wear coefficient and coefficient of friction instead of friction coefficient.
  • ANSWER// We consider the reviewer's comment to be correct and the corresponding modification was made within the entire document

Materials and Methods:

  1. Describe in which bath you did substrates-cleaning via ultrasonic?
  • ANSWER// We acknowledge the reviewer for his comment. In order to clarify the issue, the description of the ultrasonic bath used was included: “Before deposition, the substrates underwent ultrasonic cleaning for 15 minutes in alcohol, using a 45 KHz sweep frequency in a Elmasonic X-tra US 550 ultrasonic bath”
  1. (Page no. 3, lines 116) Do you mean duty cycle instead of “live time”?
  • ANSWER// We acknowledge the reviewer for his comment. The comment is correct, the live time was modified by duty cycle.

Results section:

  1. Could you use chemical composition instead of “elemental chemical composition” through the manuscript?
  • ANSWER// We acknowledge the reviewer for his comment. The statement by chemical composition was modified throughout the document.
  1. Could you use compressive stress instead of “residual stress” through the manuscript?
  • ANSWER// We acknowledge the reviewer for his comment. The statement by chemical composition was modified throughout the document.
  1. (Page no. 4, line 159) Why the authors claim that increasing the Ts leads to increase the compressive stress while there is no increase in the shifting-value of XRD peaks with increasing Ts?
  • ANSWER// We appreciate the reviewer's comment and in order to clarify the issue, figure 1b was included, where the zoom of the peaks (111) is observed for the different temperatures. Probably because the three diffraction patterns were plotted in the form of a waterfall, perspective is lost and therefore it cannot be observed clearly. However, in this image the shift towards higher diffraction angles is clearly observed.

Figure 1. a) X-ray diffraction of coatings deposited at Ts 200, 400 and 600 °C, b) zoom of the peaks (111) for the different temperatures.

  1. Regarding hardness measurements, it is recommended to employ a load in the range of 10-20 mN. Additionally, it is important to ensure that the indentation depth is less than or close to 10% of the film thickness to obtain accurate hardness values.
  • ANSWER// We appreciate the reviewer's comment and consider it very convenient. In order to clarify the issue, indentations were made with low loads basically due to two fundamental aspects, initially, due to the hardness of the material, which is considerably high, it is recommended to use low loads, this in order to reduce the stresses concentration on the surface and avoid the formation of cracks, which will be reflected in the load-displacement curve as discontinuities or pop in. This is possible because the indenter is sharp (Berkovich tip with an angle of 142° with a 10 nm radius). The second factor to take into account is the thickness of the coating. As the reviewer argues, loads that generate maximum penetrations of less than 10% of the total thickness of the coating should be used, this avoids an effect of the substrate. Which in our case happens, because the maximum penetrations were obtained for the coating deposited at 200°C, with a penetration of about 33 nm.
  1. The adhesion measurement can be influenced by several factors, including the hardness, toughness, and thickness of the films. These effects should be discussed in the manuscript to provide a comprehensive analysis of the adhesion properties.
  • ANSWER// We appreciate the reviewer's comment and consider it very convenient. To this end, some modifications were made to the adhesion analysis, taking into account the influence mainly of the mechanical properties. Additionally, the paragraph was included with their respective references:

"The observed increase in hardness (Figure 3b) would indicate an improvement in adhesion, since a harder coating tends to have improved adhesion as it is less prone to being scratched or removed from the substrate. However, the slight reduction in elastic modulus (Figure 3b) and increase in plastic deformation resistance (Figure 3c) generally leads to a reduction in resistance to crack initiation and propagation to the film/substrate interface [27, 32, 41]. "

  1. In Figure 4, it would be helpful to zoom out the image of lateral and conformal cracks to improve their visibility and distinguishability. This would enhance the clarity of the results and facilitate a better understanding of the crack morphology.
  • ANSWER// we appreciate the reviewer's comment and consider it very convenient. In order to improve the analysis and overview of scratching, an overview of each scratch mark was included in figure 4 and its description was modified in the text of the paper:

“Figure 4 shows an overview of the scratch tracks and SEM images of the cohesive (Lc1) and adhesive (Lc2) failures for the different substrate temperatures.”

Figure 4. Overview of the scratch tracks and SEM images of the failures on the critical load obtained from the scratch test for the coatings deposited at different Ts.

  1. Is it correct to describe the pinhole/droplets sites as delamination in Fig.5?
  • ANSWER// We appreciate the reviewer's comment regarding this matter. Within the indentation, distinct segments detached of the coating are observed. While these segments could potentially correspond to pores or pinholes formed on the surface during the deposition process, the geometry and positioning of these defects suggest film delamination. This is especially evident as these defects are larger compared to those observed on the surface without indentation. However, considering the possibility that these delaminations could arise from pores or pinholes generated during the deposition process, the phrase "and may come from pores or pinholes generated during deposition" has been included.

Conclusion section:

  1. The conclusion should be modified to clearly address the research question and highlight the novelty and significance of the study.
  • ANSWER// We appreciate the valuable comment provided by the reviewer and acknowledge the importance of enhancing the quality of the paper. In light of this, we have included a conclusion that highlights the significance of the obtained results in relation to the objective of the investigation. We particularly emphasize the primary application of the findings in the field of cutting tools:

“The tribological properties of TiNbN coatings, deposited using cathodic arc, were investigated under severe wear conditions simulating cutting environments. The temperature of the substrate was varied during the study. The coatings exhibited friction coefficients ranging from 0.45 to 0.8, and wear rates between 0.08 and 0.23 x10-4 mm3/Nm were observed. These values were lower than those observed for the AISI D2 steel substrate. Additionally, a quasi-stable friction stage was observed after approximately 1000 cycles. The wear processes observed on the surface of the TiNbN coatings were primarily associated with abrasion mechanisms, particularly involving third-body interactions (scratching and ploughing) and tribo-oxidation. These mechanisms are highly beneficial for cutting tools, as the formation of hard particles enhances material removal capacity, while the formation of stable oxides improves the thermal stability of the tool.”

We hope these changes are satisfactory and look forward to your feedback.

Thank you for your attention

Sincerely,

Dr. Jeferson Piamba Jimenez

Professor

Universidad de Ibague

Reviewer 2 Report

The work is interesting, well done, but there are a number of comments.

1) Lines 19-20. Ts – substrate temperature. Lines 91-92. Ts – deposition temperature. Which of these is true?

2) Table 1. Were the coating conditions selected by you earlier or taken from the literature?

3) How was the substrate heated during the coating process? This is not in the text of the manuscript.

4) Is the temperature of 600 °C too high for a cutting tool?

5) Table 4. What is the reason for such a sharp decrease in roughness on going from 200 to 400 °C?

6) Lines 225-226. "However, increasing the deposition temperature reduces the maximum penetration of the indenter, which implies an increase in hardness." But in Fig. 3b, such an increase is not noticeable.

7) Fig. 8. The caption for the figure is incorrect. Fix it.

8) The "Discussion" section is very small. I propose to merge sections 3 and 4 into one "Results and Discussion".

9) It would be good to add literature data on TiN coating in order to understand whether the addition of niobium is effective or not.

Author Response

Jeferson Piamba Jimenez

Professor

Universidad de Ibague

Carrera 22 Calle 67 B, Av. Ambala, Ibagué.

Tolima, Colombia

[email protected]

June 13, 2023

Response to Reviewer 2

Dear Reviewer 2,

First I want to thank you for taking the time to review the paper titled "EFFECT OF SUBSTRATE TEMPERATURE ON ADHESION AND TRIBOLOGICAL PROPERTIES OF TiNbN COATINGS DEPOSITED BY CATHODIC ARC" and making your constructive and highly relevant comments. All the reviews were answered and we put them to your consideration for the publication of the paper.

Below I explicitly describe each of the changes based on feedback:

The work is interesting, well done, but there are a number of comments.

1) Lines 19-20. Ts – substrate temperature. Lines 91-92. Ts – deposition temperature. Which of these is true?

  • ANSWER// We appreciate the reviewer's comment and consider it very convenient. In order to clarify the issue, changes were made throughout the document, changing the expression and leaving "substrate temperature - Ts."

2) Table 1. Were the coating conditions selected by you earlier or taken from the literature?

  • ANSWER// We appreciate the reviewer's comment. The deposit parameters have been developed through different experimental designs carried out internally by our research group. These include the optimization of the deposition rate by varying the flow of gases and the power at the cathodes, the variation of the bias voltage and, as in this case, the temperature of the substrate.

Similar parameters can be observed at: https://doi.org/10.1016/j.surfcoat.2022.128974, https://doi.org/10.1016/j.surfcoat.2021.127516, https://doi.org/10.1016/j .surfcoat.2020.125845, https://doi.org/10.1016/j.surfcoat.2019.01.070.

3) How was the substrate heated during the coating process? This is not in the text of the manuscript.

  • ANSWER// We appreciate the reviewer's comment and consider it very convenient. In order to clarify the issue, a paragraph was included in the experimental details, where the heating procedure is explained: "The substrates were heated using resistors connected to a 6kW source, and the temperature was controlled using an Impac 140 infrared pyrometer and three thermocouples located in the sample holder. The deposition process started once the temperature stabilized at the desired value for 60 minutes."

4) Is the temperature of 600 °C too high for a cutting tool?

  • ANSWER// We appreciate the reviewer's comment. Expanding the discussion in terms of substrate temperature, a temperature of 600°C is at the upper limit of use in depositing techniques, because most metallic substrates, especially ferrous ones, have tempering temperatures below 600°C. For this reason, using deposit or substrate temperatures that are higher than this value would rule out the use of most metallic substrates, since it would interfere with the thermal treatment of the material. However, specifically for the D2 steel used as a substrate, it is possible to use it for the deposit of coatings at these temperatures, since the material has good chemical stability and its quenching temperature is above 1000°C. The material can suffer from tempering at a temperature close to 600°C, however, the low exposure time (<2 hours) ensures that the thermal state of the material does not undergo major changes. Where a problem can arise and which is discussed in the paper, it is due to the thermal expansion of the material, which generates stress at the interface with the coating and reduces adhesion, despite the fact that the mechanical properties of the film increase with the temperature.

Regarding the cutting tools, there are different reports on simulation and in-situ measurements of situations where a cutting tool works on a softer material. For example, temperatures for the flank are reported to be in the range of 700 to 800°C in lubricated tests. For tests without lubrication, there are maximum points above 1000°C, however these are flash points that rapidly reduce their temperature, which quickly dissipates between the tool, the chip and the machined part (see for example https: //doi.org/10.1007/978-3-642-20617-7_6412). For these reasons, we consider that a temperature of 600°C is within the working range of the material, both in the coating production process and in the cutting process.

5) Table 4. What is the reason for such a sharp decrease in roughness on going from 200 to 400 °C?

  • ANSWER// We appreciate the reviewer's comment. The reduction in roughness is attributed to the fact that increasing the temperature of the substrate increases the mobility of reactive species on the surface, which allows them to locate and nucleate in higher energy sites. This generates coatings with higher density that in general will have less roughness. Despite the fact that the reduction observed is sharp, this trend has been also observed by other authors in the same material as a function of substrate temperature and deposition pressure (see: https://doi.org/10.1016/j. matchphys.2018.12.046).

6) Lines 225-226. "However, increasing the deposition temperature reduces the maximum penetration of the indenter, which implies an increase in hardness." But in Fig. 3b, such an increase is not noticeable.

  • ANSWER// We appreciate the reviewer's comment. In order to verify the issue, we clarify that the Oliver and Pharr model indicates that hardness is defined as the relationship between the maximum load and the contact area. According to the geometry of the indenter, the contact area is a function of the maximum contact depth. In such a way that at lower maximum penetration depths, the contact area will be smaller and therefore higher hardness values. See (W. C. Oliver and G. M. Pharr: An improved technique for determining hardness and elastic modulus. J. Mater. Res., Vol. 7, No. 6, June 1992). In addition to this, figure 3b shows a slight increase in hardness, which corresponds to an increase of around 11%, which is attributed to the increase in residual compressive stresses generated by the differences in coefficient of thermal expansion between the film and the substrate and the difference in atomic radii of Ti and Nb.

7) Fig. 8. The caption for the figure is incorrect. Fix it.

  • ANSWER// We appreciate the reviewer's comment. The description of figure 8 was modified by: "Wear rate for the D2 steel substrate and the coatings deposited at different temperatures."

8) The "Discussion" section is very small. I propose to merge sections 3 and 4 into one "Results and Discussion".

  • ANSWER// We appreciate the reviewer's comment and consider it correct. For this reason, sections 3 and 4 were unified, leaving a single section "3. Results and discussion" and section 4 was assigned to conclusions.

9) It would be good to add literature data on TiN coating in order to understand whether the addition of niobium is effective or not.

  • ANSWER// We appreciate the reviewer's comment and consider it important to broaden the discussion in terms of the increase in TiN properties by adding different elements to the crystalline structure, in this case Nb. To address this, we have included additional paragraphs in the Results and Discussion section, which specifically focus on the influence of process parameters in relation to the approximation of cutting conditions and their impact on the tribological properties:

“Various researchers have conducted investigations on the tribological properties of TiN under conditions that simulate severe wear scenarios resembling those encountered in cutting tools. These studies have revealed that the friction coefficients of TiN can range between 0.3 and 0.8, depending on the specific contact conditions. When subjected to high loads and moderate speeds (<50 cm/sec), the coefficient of friction tends to decrease due to the formation of lubricating oxide layers. However, wear rates increase significantly, surpassing 30 x 10-4 mm3/Nm [56-59]. Furthermore, it has been observed that the dominant wear mechanisms for TiN coatings include abrasion, scratching, and ploughing caused by hard particles. Nevertheless, most of the references available primarily focus on coatings with low thicknesses (<2 μm), resulting in relatively short lifetimes of less than 2000 cycles. After the coating undergoes wear, which may be partial, both the coefficient of friction and the wear rate tend to rise, reaching values as high as 0.9 and 40 x 10-4 mm3/Nm, respectively [56, 57, 59]. Similar-ly, studies have demonstrated that incorporating elements that stabilize oxide formation, such as Al in TixAlyN [59, 60], Hf in TixHfyN [9], and Nb in TiNbN [25], can lead to reductions in the coefficient of friction and abrasion at both low and high temperatures. These modifications have resulted in an extended useful life for the surface, ultimately enhancing the material's overall performance in applications that include cutting tools, cold and hot forming and drawing, erosion and impact [60, 61].”

We hope these changes are satisfactory and look forward to your feedback.

Thank you for your attention

Sincerely,

Dr. Jeferson Piamba Jimenez

Professor

Universidad de Ibague

Reviewer 3 Report

The manuscript investigates the influence of substrate temperature (Ts) on the adhesion and tribological properties of niobium-doped titanium nitride (TiNbN) coatings deposited on D2 steel substrates using cathodic arc deposition. The study utilized X-ray diffraction, nanoindentation, adhesion analysis, and pin-on-disk tests to assess the properties of the coatings. X-ray diffraction confirmed the presence of TiNbN coatings with a face-centered cubic (FCC) crystalline structure, where niobium atoms substituted titanium atoms in the TiN lattice. The lattice parameter decreased and the crystallite shape transitioned from flat-like to sphere-like with increasing substrate temperature.

In industrial manufacturing, coating is a widely-used method to improve the tribological properties. More recent works about the application of different coating techniques on D2 or equivalent steels should be introduced, such as: doi.org/10.3390/coatings12060793; doi.org/10.1115/1.4050902.

Nanoindentation tests revealed that the hardness (H) of the coatings increased with substrate temperature, while the elasticity modulus (E) decreased. This resulted in improved elastic strain limit for failure (H/E) and enhanced plastic deformation resistance (H3/E2), indicating increased stiffness and contact elasticity.

Adhesion was evaluated through critical load measurements. The coatings exhibited higher critical loads (~50N) at substrate temperatures of 200°C and 400°C compared to a lower critical load (~38N) at 600°C. Cohesive failures, characterized by lateral cracking, were observed, as well as adhesive failures attributed to chipping spallation.

The tribological properties were assessed using a pin-on-disk test. Friction coefficients increased with substrate temperature, although they remained lower than those of the substrate material. The surface morphology influenced friction and wear behavior, facilitating the formation of abrasive particles. However, the absence of coating detachment in the wear tracks indicated that the TiNbN coatings were capable of withstanding load and wear.

Overall, the study highlights the influence of substrate temperature on the adhesion and tribological properties of TiNbN coatings deposited by cathodic arc. The findings demonstrate the potential of these coatings to enhance hardness, elastic strain limit, and plastic deformation resistance. The results also indicate that the coatings exhibit good adhesion and wear resistance, making them suitable for various applications where improved adhesion and tribological performance are desired.

The quality of English language in the provided manuscript is quite good. The language is clear, concise, and effectively communicates the key information and findings of the study.

Author Response

Jeferson Piamba Jimenez

Professor

Universidad de Ibague

Carrera 22 Calle 67 B, Av. Ambala, Ibagué.

Tolima, Colombia

[email protected]

June 13, 2023

Response to Reviewer 3

Dear Reviewer 3,

First I want to thank you for taking the time to review the paper titled "EFFECT OF SUBSTRATE TEMPERATURE ON ADHESION AND TRIBOLOGICAL PROPERTIES OF TiNbN COATINGS DEPOSITED BY CATHODIC ARC" and making your constructive and highly relevant comments. All the reviews were answered and we put them to your consideration for the publication of the paper.

Below I explicitly describe each of the changes based on feedback:

The manuscript investigates the influence of substrate temperature (Ts) on the adhesion and tribological properties of niobium-doped titanium nitride (TiNbN) coatings deposited on D2 steel substrates using cathodic arc deposition. The study utilized X-ray diffraction, nanoindentation, adhesion analysis, and pin-on-disk tests to assess the properties of the coatings. X-ray diffraction confirmed the presence of TiNbN coatings with a face-centered cubic (FCC) crystalline structure, where niobium atoms substituted titanium atoms in the TiN lattice. The lattice parameter decreased and the crystallite shape transitioned from flat-like to sphere-like with increasing substrate temperature.

  • ANSWER// We appreciate the reviewer's comment. In order to improve the analysis, figure 1b was included, where the zoom of the peaks (111) is observed for the different temperatures.

Figure 1. a) X-ray diffraction of coatings deposited at Ts 200, 400 and 600 °C, b) zoom of the peaks (111) for the different temperatures.

In industrial manufacturing, coating is a widely-used method to improve the tribological properties. More recent works about the application of different coating techniques on D2 or equivalent steels should be introduced, such as: doi.org/10.3390/coatings12060793; doi.org/10.1115/1.4050902.

  • ANSWER// We appreciate the reviewer's comment and consider it important to broaden the discussion in terms of the increase in different coatings properties by adding different elements to the crystalline structure, in this case Nb, the use of different substrates similar to the AISI D2 steel and several applications of the coated surfaces. To address this, we have included additional paragraphs in the Results and Discussion section, which specifically focus on the influence of process parameters in relation to the approximation of cutting tools, cold and hot forming and drawing, erosion and impact and their impact on the tribological properties:

“Various researchers have conducted investigations on the tribological properties of TiN under conditions that simulate severe wear scenarios resembling those encountered in cutting tools. These studies have revealed that the friction coefficients of TiN can range between 0.3 and 0.8, depending on the specific contact conditions. When subjected to high loads and moderate speeds (<50 cm/sec), the coefficient of friction tends to decrease due to the formation of lubricating oxide layers. However, wear rates increase significantly, surpassing 30 x 10-4 mm3/Nm [56-59]. Furthermore, it has been observed that the dominant wear mechanisms for TiN coatings include abrasion, scratching, and ploughing caused by hard particles. Nevertheless, most of the references available primarily focus on coatings with low thicknesses (<2 μm), resulting in relatively short lifetimes of less than 2000 cycles. After the coating undergoes wear, which may be partial, both the coefficient of friction and the wear rate tend to rise, reaching values as high as 0.9 and 40 x 10-4 mm3/Nm, respectively [56, 57, 59]. Similarly, studies have demonstrated that incorporating elements that stabilize oxide formation, such as Al in TixAlyN [59, 60], Hf in TixHfyN [9], and Nb in TiNbN [25], can lead to reductions in the coefficient of friction and abrasion at both low and high temperatures. These modifications have resulted in an extended useful life for the surface, ultimately enhancing the material's overall performance in applications that include cutting tools, cold and hot forming and drawing, erosion and impact [60, 61].”

Nanoindentation tests revealed that the hardness (H) of the coatings increased with substrate temperature, while the elasticity modulus (E) decreased. This resulted in improved elastic strain limit for failure (H/E) and enhanced plastic deformation resistance (H3/E2), indicating increased stiffness and contact elasticity.

  • ANSWER// We acknowledge the reviewer for his comment and hope that the small changes made in the mechanical properties section will improve the understanding of the analysis performed and contribute to the improvement of the paper.

Adhesion was evaluated through critical load measurements. The coatings exhibited higher critical loads (~50N) at substrate temperatures of 200°C and 400°C compared to a lower critical load (~38N) at 600°C. Cohesive failures, characterized by lateral cracking, were observed, as well as adhesive failures attributed to chipping spallation.

  • ANSWER// We appreciate the reviewer's comment. Modifications were made that we hope will improve the quality of the paper. A paragraph was included where it analyses the influence of the mechanical properties on the adhesion of the coatings:

"The observed increase in hardness (Figure 3b) would indicate an improvement in adhesion, since a harder coating tends to have improved adhesion as it is less prone to being scratched or removed from the substrate. However, the slight reduction in elastic modulus (Figure 3b) and increase in plastic deformation resistance (Figure 3c) generally leads to a reduction in resistance to crack initiation and propagation to the film/substrate interface [25, 30, 37]. "

Figure 4 was modified, including general images of the scratch marks and the areas where the different adhesion failures occurred:

Figure 4. Overview of the scratch tracks and SEM images of the failures on the critical load obtained from the scratch test for the coatings deposited at different Ts.

The tribological properties were assessed using a pin-on-disk test. Friction coefficients increased with substrate temperature, although they remained lower than those of the substrate material. The surface morphology influenced friction and wear behavior, facilitating the formation of abrasive particles. However, the absence of coating detachment in the wear tracks indicated that the TiNbN coatings were capable of withstanding load and wear.

  • ANSWER// We appreciate the reviewer's comment. Changes were made to the tribological properties section in order to improve the analysis performed.

The caption of figure 8 was modified: “Wear rate for the D2 steel substrate and the coatings deposited at different temperatures.”

A paragraph was included where the permissibility of the results obtained is analyzed, the reference framework was expanded, including an analysis of the base material and applications related to cutting tools:

“Various researchers have conducted investigations on the tribological properties of TiN under conditions that simulate severe wear scenarios resembling those encountered in cutting tools. These studies have revealed that the friction coefficients of TiN can range between 0.3 and 0.8, depending on the specific contact conditions. When subjected to high loads and moderate speeds (<50 cm/sec), the coefficient of friction tends to decrease due to the formation of lubricating oxide layers. However, wear rates increase significantly, surpassing 30 x 10-4 mm3/Nm [56-59]. Furthermore, it has been observed that the dominant wear mechanisms for TiN coatings include abrasion, scratching, and ploughing caused by hard particles. Nevertheless, most of the references available primarily focus on coatings with low thicknesses (<2 μm), resulting in relatively short lifetimes of less than 2000 cycles. After the coating undergoes wear, which may be partial, both the coefficient of friction and the wear rate tend to rise, reaching values as high as 0.9 and 40 x 10-4 mm3/Nm, respectively [56, 57, 59]. Similar-ly, studies have demonstrated that incorporating elements that stabilize oxide formation, such as Al in TixAlyN [59, 60], Hf in TixHfyN [9], and Nb in TiNbN [25], can lead to reductions in the coefficient of friction and abrasion at both low and high temperatures. These modifications have resulted in an extended useful life for the surface, ultimately enhancing the material's overall performance in applications that include cutting tools, cold and hot forming and drawing, erosion and impact [60, 61].”

Overall, the study highlights the influence of substrate temperature on the adhesion and tribological properties of TiNbN coatings deposited by cathodic arc. The findings demonstrate the potential of these coatings to enhance hardness, elastic strain limit, and plastic deformation resistance. The results also indicate that the coatings exhibit good adhesion and wear resistance, making them suitable for various applications where improved adhesion and tribological performance are desired.

  • ANSWER// We appreciate the reviewer's comment and we hope that the modifications made enhance the scope and the research problem addressed, explaining the results obtained and the conclusions in greater depth. In the conclusions section, a paragraph detailing the general conclusion of the paper was included:

“The tribological properties of TiNbN coatings, deposited using cathodic arc, were investigated under severe wear conditions simulating cutting environments. The tem-perature of the substrate was varied during the study. The coatings exhibited friction coefficients ranging from 0.45 to 0.8, and wear rates between 0.08 and 0.23 x10-4 mm3/Nm were observed. These values were lower than those observed for the AISI D2 steel substrate. Additionally, a quasi-stable friction stage was observed after approxi-mately 1000 cycles. The wear processes observed on the surface of the TiNbN coatings were primarily associated with abrasion mechanisms, particularly involving third-body interactions (scratching and ploughing) and tribo-oxidation. These mecha-nisms are highly beneficial for cutting tools, as the formation of hard particles enhanc-es material removal capacity, while the formation of stable oxides improves the ther-mal stability of the tool.”

We hope these changes are satisfactory and look forward to your feedback.

Thank you for your attention

Sincerely,

Dr. Jeferson Piamba Jimenez

Professor

Universidad de Ibague

Round 2

Reviewer 1 Report

Thank you for considering the suggested modification.

The following suggestion aims to provide a clearer and more descriptive representation of the study:

The proposed title can be modified as follows: "TiNbN hard coating deposited at varied substrate temperature by cathodic arc: Tribological performance under simulated cutting conditions"

The authors are encouraged to enhance the clarity and readability of the English language used in the manuscript.

Author Response

Response to Reviewer 1

Dear Reviewer 1,

Regards, Again we want to acknowledge the reviewer for taking the time to review the paper and making your constructive comments.

Below I explicitly describe each of the changes based on feedback:

The proposed title can be modified as follows: "TiNbN hard coating deposited at varied substrate temperature by cathodic arc: Tribological performance under simulated cutting conditions".

  • ANSWER// We appreciate the suggestion and consider it correct, for this reason, the title of the paper was modified " TiNbN hard coating deposited at varied substrate temperature by cathodic arc: Tribological performance under simulated cut-ting conditions"

Comments on the Quality of English Language

The authors are encouraged to enhance the clarity and readability of the English language used in the manuscript.

  • ANSWER// We appreciate the suggestion and inform you that the text was revised to improve the style and grammar.

We hope these changes are satisfactory and look forward to your feedback.

Thank you for your attention

Sincerely,

Dr. Jeferson Piamba Jimenez

Professor

Universidad de Ibague

Reviewer 2 Report

The manuscript can be accepted for publication.

Author Response

Dear Reviewer 2,

Regards,

Again we want to acknowledge the reviewer for taking the time to review the paper and making your constructive comments.

Below I explicitly describe each of the changes based on feedback:

Comments on the Quality of English Language

The authors are encouraged to enhance the clarity and readability of the English language used in the manuscript.

  • ANSWER// We appreciate the suggestion and inform that the text was revised in order to improve the style and grammar.

We hope these changes are satisfactory and look forward to your feedback.

Thank you for your attention

Sincerely,

Dr. Jeferson Piamba Jimenez

Professor

Universidad de Ibague
